# Designing a Conditional Prior Distribution for Flow-Based Generative Models

**Noam Issachar**\*  
*The Hebrew University of Jerusalem*

*noam.issachar@mail.huji.ac.il*

**Mohammad Salama**\*  
*The Hebrew University of Jerusalem*

*mohammad.salama3@mail.huji.ac.il*

**Raanan Fattal**  
*The Hebrew University of Jerusalem*

*raanan.fattal@mail.huji.ac.il*

**Sagie Benaim**  
*The Hebrew University of Jerusalem*

*sagie.benaim@mail.huji.ac.il*

**Reviewed on OpenReview:** *https: // openreview. net/ forum? id= Teh9Bq4giF*

## Abstract

Flow-based generative models have recently shown impressive performance for conditional generation tasks, such as text-to-image generation. However, current methods transform a general unimodal noise distribution to a specific mode of the target data distribution. As such, every point in the initial source distribution can be mapped to every point in the target distribution, resulting in long average paths. To this end, in this work, we tap into a non-utilized property of conditional flow-based models: the ability to design a non-trivial prior distribution. Given an input condition, such as a text prompt, we first map it to a point lying in data space, representing an "average" data point with the minimal average distance to all data points of the same conditional mode (e.g., class). We then utilize the flow matching formulation to map samples from a parametric distribution centered around this point to the conditional target distribution. Experimentally, our method significantly improves training times and generation efficiency (FID, KID and CLIP alignment scores) compared to baselines, producing high quality samples using fewer sampling steps. Code is available at `https://github.com/MoSalama98/conditional-prior-flow-matching`.

## 1 Introduction

Conditional generative models are of significant importance for many scientific and industrial applications. Of these, the class of flow-based models and score-based diffusion models has recently shown a particularly impressive performance (27; 11; 9; 18). Although impressive, current methods suffer from long training and sampling times. To this end, in this work, we tap into a non-utilized property of conditional flow-based models: the ability to design a non-trivial prior distribution for conditional flow-based models based on the input condition. In particular, for class-conditional generation and text-to-image generation, we propose a *robust* method for constructing a conditional flow-based generative model using an informative condition-specific prior distribution fitted to the conditional modes (e.g., classes) of the target distribution. By better modeling the prior distribution, we aim to improve the efficiency, both at training and at inference, of conditional generation via flow matching, thus achieving high quality results with fewer sampling steps.

Given an input variable (e.g., a class or text prompt), current flow-based and score-based diffusion models combine the input condition with intermediate representations in a learnable manner. However, crucially,

---

\*Equal contribution

these models are still trained to transform a generic unimodal noise distribution to the different modes of the target data distribution. In some formulations, such as score-based diffusion (16; 38; 37), the use of a Gaussian source density is intrinsically connected to the process constructing the transformation. In others, such as flow matching (27; 29; 1), a Gaussian source is not required, but is often chosen as a default for convenience. Consequently, in these settings, the prior distribution bears little or no resemblance to the target, and hence every point in the initial source distribution can be mapped to every point in every mode in the target distribution, corresponding to a given condition. This means that the average distance between pairs of source-target points is fairly large.

In the unconditional setting, recent works (33; 41), show that starting from a source (noise) data point that is close to the target data sample, during training, results in straighter probability flow lines, fewer sampling steps at test time, and faster training time. This is in comparison to the non-specific random pairing between the distributions typically used for training flow-based and score-based models. This suggests that finding a strategy to minimize the average distance between source and target points could result in a similar benefit. Our work aims to construct this by constructing a *condition-specific* source distribution by leveraging the input condition.

We, therefore, propose a novel paradigm for designing an informative condition-specific prior distribution for a flow-based conditional generative model. While in this work, we choose to work on *flow matching*, our approach can also be incorporated in other generative models, supporting arbitrary prior distributions. In the first step, we embed the input condition $c$ to a point $x_c$ lying in data space (which can be a latent one). For a discrete set of classes, this is done by averaging training samples corresponding to a given class $c$ in the data space. In the continuous case, such as text-to-image, we first choose a meaningful embedding for the input condition $c$ (*e.g.* CLIP (34)). Given a training sample $x_c$ and the corresponding conditional embedding $e_c$, we train a deterministic mapper function that projects $e_c$ to $x_c$ lying in data space. This results in an "average" data point of all samples $x$ corresponding to the condition $c$. To enable stochastic mapping, we then map samples from a parametric conditional distribution centered on $x_c$ to the conditional target distribution $\rho_1(x|c)$.

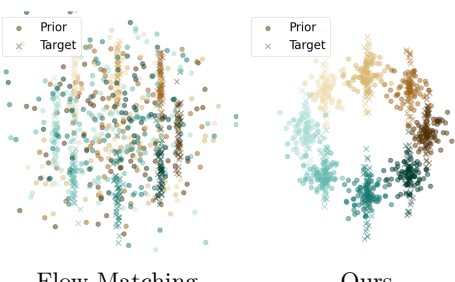

Flow Matching        Ours

Figure 1: **Illustration of our approach.** The LHS illustrates the flow matching paradigm, where every sample in the source Gaussian (shown as a circular point) can be mapped to every sample in the conditional target mode (shown as a cross point), where each class samples are shown in a different color. In contrast, our method, shown on the RHS, constructs a class-specific conditional distribution as a source prior distribution. Each sample in the source distribution is, on average, closer to its corresponding sample in the target mode.

While our approach can be implemented with any parametric conditional distribution, in our experiments, we chose to utilize a Gaussian Mixture Model (GMM). Specifically, the mean of each Gaussian is the "average" conditional data point. In the discrete condition case, each prior-Gaussian's covariance is estimated directly from the class-dependent training data, while for the continuous setting, it is fixed as a hyperparameter. Further, while the data space can be arbitrary, we choose it as the latent space of a pre-trained variational autoencoder (VAE). These choices are derived from the following desirable properties: (i). One can easily sample from a GMM, (ii). Class conditional information can be directly represented by a GMM, with each Gaussian corresponding to a conditional mode. (iii). We find empirically (see Sec. 4.2), for real-world distributions (ImageNet (8) and MS-COCO (26)), that the average distance between pairs of samples from the prior and data distributions (*i.e.* the *transport cost*) is much smaller than the unimodal Gaussian alternative (as in (27; 33)). Moreover, previous works (22; 4; 15) has shown that applying a GMM in a VAE space can be highly effective for clustering, suggesting that it can act as a suitable prior distribution. An illustration of our approach, for a simple setting consisting of eight Gaussians, each representing a different class, is shown in Fig. 1.

To validate our approach, we first formulate flow matching from our conditional prior distribution (CPD) and show that our formulation results in low global truncation errors. Next, we consider a toy setting with

a known analytical target distribution and illustrate our method's advantage in efficiency and quality. For real-world datasets, we consider both the MS-COCO (text-to-image generation) and ImageNet-64 datasets (class conditioned generation). Compared to other flow-based (CondOT (27), BatchOT (33)) and diffusion (DDPM (17)) based models, our approach allows for faster training and sampling, as well as for a significantly improved generated image quality and diversity, evaluated using FID and KID, and alignment to the input text, evaluated using CLIP score.

## 2 Related Work

**Flow-based Models.** Continuous Normalizing Flows (CNFs) (6) emerged as a novel paradigm in generative modeling, offering a continuous-time extension to the discrete Normalizing Flows (NF) framework (24; 32). Recently, Flow Matching (27; 29; 1) has been introduced as a simulation-free alternative for training CNFs. In scenarios involving conditional data (e.g., in text-to-image generation), conditioning is applied similarly to diffusion models, often through cross attention between the input condition and latent features. Typically, the source distribution remains unimodal, like a standard Gaussian (28). In contrast, our approach derives a prior distribution that is dependent on the input condition.

**Conditioning in Flow-based Models.** Conditional generation often adapts diffusion techniques, leveraging the connection between diffusion and probability flow ODEs (e.g., (39). This typically involves parameterizing $v(t, x, c)$ via Classifier Free Guidance inspired approaches (18; 7; 45), vector field differences (19), or cross-attention, usually mapping from a fixed Gaussian prior. Our contribution—designing an adaptive, condition-specific prior—is *orthogonal* to these dynamics conditioning techniques. We leverage such standard mechanisms (specifically, cross-attention, as in our baselines) and apply our novel prior in addition.

**Informative Prior Design.** Designing useful priors is studied in generative models like VAEs (e.g., (10)) and Normalizing Flows (e.g., (21)). In score-based diffusion and flow matching, some works designed informative priors. For score-based diffusion, PriorGrad (25) addresses the well-known failure mode of score-based diffusion models to deal with extreme modes in the data, such as the voiced and unvoiced parts in waveform signals. This method whitens the data by normalizing mode-specific statistics, effectively flattening these modes so that the diffusion process operates on a simpler, quasi-isotropic distribution. This strategy reduces the burden on the network but also suppresses much of the original signal's structure. By contrast, our approach takes the opposite stance, instead of simplifying the data, we adapt the prior to better match the data distribution, thereby preserving its multimodal complexity while still enabling stable training. More recently, for flow matching, (33; 41) utilized dynamic optimal transport (OT) across mini-batches to construct priors. Despite enabling efficient sampling, these OT-based methods face challenges with highly expensive training (quadratic OT computation) and reduced effectiveness for high-dimensional data, requiring exponentially larger batch sizes for performance gains. Our approach avoids these specific limitations by leveraging the conditioning variable to directly shape the prior distribution.

## 3 Preliminaries

We begin by introducing Continuous Normalizing Flow (6) in Sec. 3.1 and Flow Matching (27) in Sec. 3.2. This will motivate our approach, detailed in Sec. 4, which defines an informative conditional prior distribution on a conditional flow model.

### 3.1 Continuous Normalizing Flows

A probability density function over a manifold $\mathcal{M}$ is a continuous non-negative function $\rho : \mathcal{M} \to \mathbb{R}_+$ such that $\int \rho(x) dx = 1$. We set $\mathcal{P}$ to be the space of such probability densities on $\mathcal{M}$. A *probability path* $\rho_t : [0, 1] \to \mathcal{P}$ is a curve in probability space connecting two densities $\rho_0, \rho_1 \in \mathcal{P}$ at endpoints $t = 0, t = 1$. A *flow* $\psi_t : [0, 1] \times \mathcal{M} \to \mathcal{M}$ is a time-dependent diffeomorphism defined to be the solution to the Ordinary Differential Equation (ODE):

$$\frac{d}{dt}\psi_t(x) = u_t\left(\psi_t(x)\right), \quad \psi_0(x) = x \tag{1}$$

subject to initial conditions where $u_t : [0, 1] \times \mathcal{M} \to \mathcal{T}\mathcal{M}$ is a time-dependent smooth vector field on the collection of all tangent planes on the manifold $\mathcal{T}\mathcal{M}$ (*tangent bundle*). A flow $\psi_t$ is said to generate a probability path $\rho_t$ if it 'pushes' $\rho_0$ forward to $\rho_1$ following the time-dependent vector field $u_t$. The path is denoted by:

$$\rho_t = [\psi_t]_\# \rho_0 := \rho_0(\psi_t^{-1}(x)) \det \left| \frac{d\psi_t^{-1}}{dx}(x) \right| \tag{2}$$

where $\#$ is the standard push-forward operation. Previously, (6) proposed to model the flow $\psi_t$ implicitly by parameterizing the vector field $u_t$, to produce $\rho_t$, in a method called *Continuous Normalizing Flows* (CNF).

### 3.2 Flow Matching

Flow Matching (FM) (27) is a simulation-free method for training CNFs that avoids likelihood computation during training, which can be expensive. It does so by fitting a vector field $v_t^\theta$ with parameters $\theta$ and regressing vector fields $u_t$ that are known *a priori* to generate a probability path $\rho_t \in \mathcal{P}$ satisfying the boundary conditions:

$$\rho_0 = p, \quad \rho_1 = q \tag{3}$$

Note that $u_t$ is generally intractable. However, a key insight of (27), is that this vector field can be constructed based on conditional vector fields $u_t(x|x_1)$ that generate conditional probability paths $\rho_t(x|x_1)$. The push-forward of the conditional flow $\psi_t(x|x_1)$, start at $\rho_t$ and concentrate the density around $x = x_1 \in \mathcal{M}$ at $t = 1$. Marginalizing over the target distribution $q$ recovers the unconditional probability path and unconditional vector field: $\rho_t(x) = \int_\mathcal{M} \rho_t(x|x_1) q(x_1) dx_1$, $u_t(x) = \int_\mathcal{M} u_t(x|x_1) \frac{\rho_t(x|x_1) q(x_1)}{\rho_t(x)} dx_1$.

This vector field can be matched by a parameterized vector field $v_\theta$ using the $\mathcal{L}_{\text{cfm}}(\theta)$ objective:

$$\mathbb{E}_{t \sim \mathcal{U}(0,1), q(x_1), \rho_t(x|x_1)} \|v_\theta(t, x) - u_t(x|x_1)\|^2 \tag{4}$$

where $\|\cdot\|$ is a norm on $\mathcal{T}\mathcal{M}$. One particular choice of a conditional probability path $\rho_t(x|x_1)$ is to use the flow corresponding the optimal transport displacement interpolant (30) between Gaussian distributions. Specifically, in the context of the conditional probability path, $\rho_0(x|x_1)$ is the standard Gaussian, a common convention in generative modeling, and $\rho_1(x|x_1)$ is a small Gaussian centered around $x_1$. The conditional flow interpolating these distributions is given by $x_t = \psi_t(x|x_1) = (1-t)x_0 + tx_1$, which results in the following conditional vector field $u_t(x|x_1) = \frac{x_1 - x}{1 - t}$, which is marginalized in Eq. 4. Substituting $x_t$ to $u_t(x|x_1)$, one can also express the value of this vector field using a simpler expression: $u_t(x_t|x_1) = x_1 - x_0$.

**Conditional Generation via Flow Matching.** Flow matching (FM) was extended to conditional generative modeling in several works (45; 7; 2; 20). In contrast to the original FM, one first samples a condition $c$. One then produces samples from $p_t(x|c)$ by passing $c$ as input to the parametric vector field $v_\theta$. The *Conditional Generative Flow Matching* (CGFM) objective $\mathcal{L}_{\text{cgfm}}(\theta)$ is:

$$\mathbb{E}_{t \sim \mathcal{U}(0,1), q(x_1, c), \rho_t(x|x_1)} \|v_\theta(t, c, x) - u_t(x|x_1)\|^2 \tag{5}$$

In practice, $c$ is incorporated by embedding it into some representation space and then using cross-attention between it and the features of $v_\theta$ as in (35).

**Flow Matching with Joint Distributions.** While (27) considered the setting of independently sampled $x_0$ and $x_1$, recently, (33; 41) generalized the FM framework to an arbitrary joint distribution of $\rho(x_0, x_1)$ in the unconditional generation setting, which satisfies the following marginal constraints: $\int \rho(x_0, x_1) dx_1 = q(x_0)$, $\int \rho(x_0, x_1) dx_0 = q(x_1)$. (33) modified the conditional probability path construction so at $t = 0$, $\rho_0(x_0|x_1) = p(x_0|x_1)$, where $p(x_0|x_1)$ is the conditional distribution $\frac{\rho(x_0, x_1)}{q(x1)}$. The *Joint Conditional Flow Matching* (JCFM) objective is:

$$\mathcal{L}_{\text{jcfm}}(\theta) = \mathbb{E}_{t \sim \mathcal{U}(0,1), \rho(x_0, x_1)} \|v_\theta(t, x) - u_t(x|x_1)\|^2 \tag{6}$$

## 4 Method

Given a set $\{x_{1_i}, c_i\}_{i=1}^m$ of input samples and their corresponding conditioning states, our goal is to construct a flow-matching model that samples from $q(x_1|c)$ that start from our conditional prior distribution (CPD).

### 4.1 Flow Matching from Conditional Prior Distribution

We generalize the framework of Sec. 3.2 to a construction that uses an arbitrary conditional joint distribution of $\rho(x_0, x_1, c)$ which satisfy the marginal constraints:

$$\int \rho(x_0, x_1, c) dx_0 = q(x_1, c), \quad \int \rho(x_0, x_1, c) dx_1 dc = p(x_0)$$

Then, building on flow matching, we propose to modify the conditional probability path so that at $t = 0$, we define:

$$\rho_0(x_0|x_1, c) = p(x_0|x_1, c) \tag{7}$$

where $p(x_0|x_1, c)$ is the conditional distribution $\frac{\rho(x_0, x_1, c)}{q(x_1, c)}$. Using this construction, we satisfy the boundary condition of Eq. 3:

$$\rho_0(x_0) = \int \rho_0(x_0|x_1, c) q(x_1, c) dx_1 dc \tag{8}$$

$$= \int p(x_0|x_1, c) dx_1 dc = p(x_0) \tag{9}$$

The conditional probability path $\rho_t(x|x_1, c)$ does not need to be explicitly formulated. Instead, only its corresponding conditional vector field $u_t(x|x_1, c)$ needs to be defined such that points $x_0$ drawn from the conditional prior distribution $\rho_0(x_0|x_1, c)$, reach $x_1$ at $t = 1$, i.e., reach distribution $\rho_1(x|x_1, c) = \delta(x - x_1)$. We thus purpose the *Conditional Generation Joint FM* $\mathcal{L}_{\text{cgjfm}}(\theta)$ objective:

$$\mathbb{E}_{t \sim \mathcal{U}(0,1), q(x_0, x_1, c)} \|v_\theta(t, x, c) - u_t(x|x_1, c)\|^2 \tag{10}$$

where $x = \psi_t(x_0|x_1, c)$. Training only involves sampling from $q(x_0, x_1, c)$ and does not require explicitly defining the densities $q(x_0, x_1, c)$ and $\rho_t(x|x_1, c)$. We note that this objective is reduced to the CGFM objective Eq. 5 when $q(x_0, x_1, c) = q(x_1, c)p(x_0)$.

### 4.2 Conditional Prior Distribution

We now describe our choice of a condition-specific prior distribution. When choosing a conditional prior distribution we want to adhere to the following design principles: (i) *Easy to sample*: can be efficiently sampled from. (ii) Well represents the target conditional modes. We design a condition-specific prior distribution based on a parametric *Mixture Model* where each mode of the mixture is correlated to a specific conditional distribution $p(x_1|c)$. Specifically, we choose the prior distribution to be the following, *easy to sample*, *Gaussian Mixture Model* (GMM):

$$p_0 = \text{GMM}(\mathcal{N}(\mu_i, \Sigma_i)_{i=1}^n, \pi) \tag{11}$$

where $\pi \in \mathbb{R}^n$ is a probability vector associated with the number of conditions $n$ (could be $\infty$) and $\mu_i, \Sigma_i$ are parameters determined by the conditional distribution $q(x_1|c_i)$ statistics, *i.e.*

$$\mu_i = \mathbb{E}[x_1|c_i], \quad \Sigma_i = \text{cov}[x_1|c_i] \tag{12}$$

To sample from the marginal distribution $p(x_0|x_1, c_i)$, we sample from the cluster $\mathcal{N}(\mu_i, \Sigma_i)$ associated with the condition $c_i$.

**Obtaining a Lower Global Truncation Error.** Our CPD fits a GMM to the data distribution in a favorable setting, where the association between samples and clusters is given. In this process, we fit a dedicated Gaussian distribution to data points with the same condition. If the latter are close to being unimodal, this approximation is expected to be tight, in terms of the average distances between samples from the condition data mode and the fitted Gaussian. Tab. 1 provides the average distances between pairs of samples from the prior and data distributions (i.e. the

Table 1: Average distances between samples from the prior and data distributions (*i.e. transport cost*) on the ImageNet-64 and MS-COCO datasets across baselines.

| | ImageNet-64 | MS-COCO |
|---|---|---|
| CondOT | 640 | 630 |
| BatchOT | 632 | 604 |
| Ours | **570** | **510** |

*transport cost*) of CondOT (27), BatchOT (33) and our CPD over the ImageNet-64 (8) and MS-COCO (26) datasets (note that both CondOT and BatchOT are applied in the conditinal setting, see Sec. 5.2.1 for details). As expected, BatchOT which minimizes this exact measure within mini-batches, obtains better scores than the naïve pairing used in CondOT, while our CPD, which approximates the data using a GMM exploits the conditioning available in these datasets, and obtains considerably lower average distances.

As noted in (33), lower transport cost is generally associated with straighter flow trajectories, more efficient sampling and lower training time. We want to substantiate this claim from the viewpoint of cumulative errors in numerical integration. Sampling from flow-based models consists of solving a time-dependent ODE of the form $\dot{x}_t = u_t(x_t)$, where $u_t$ is the velocity field. This equation is solved by the following integral $x_t = \int_0^t u_s(x_s)ds$, where the initial condition $x_0$ is sampled from the prior distribution. Numerical integration over discrete time steps accumulate an error at each step $n$ which is known as the *local truncation error $\tau_n$*, which accumulates into what is known as the *global truncation error $e_n$*. This error is bounded by  (40)

$$|e_n| \leq \frac{max_j \tau_j}{hL}\big(e^{L(t_n - t_0)} - 1\big) \tag{13}$$

where $h$ is the step size and $L$ is the Lipschitz constant of $u_t$. The distance between the endpoints of a path $\Delta = |x_1 - x_0|$ is given by $|\int_0^1 u_s(x_s)ds|$ which can be interpreted as the magnitude of the average velocity along the path $x_t$. Hence, the longer the path $\Delta$, the larger the integrated flow vector field $u_t$. For example, if we scale a path uniformly by a factor $C > 1$, i.e., $x_t \to C(x_t)$, we get, $\frac{d}{dt}C(x_t) = C(u_t)$ in which case the Lipschitz constant $L$ is also multiplied by $C$.

By shortening the distance between the prior and data distribution, as our CPD does, we lower the integration errors which permits the use of coarser integration steps, which in turn yield smaller global errors. Thus, our construction allows for fewer integration steps during sampling. In Appendix A, we also provide a theoretical justification for why we expect the transition error to decrease as the prior distribution moves closer to the data.

### 4.2.1  Construction

Next, we explain how we construct $p_0$ (Eq. 11) for both the discrete case (e.g., class conditional generation) and continuous case (e.g., text conditional generation).

**Discrete Condition.**   In the setup of discrete conditional generation, we are given data $\{x_{1_i}, c_i\}_{i=1}^m$ where there are a finite set of conditions $c_i$. We approximate the statistics of Eq. 12 using the training data statistics. That is, we compute the mean and covariance matrix of each class (potentially in some latent representation of a pretrained auto-encoder). Since the classes at inference time are the same as in training, we use the same statistics at inference.

**Continuous Condition.**   While in the discrete case we can directly approximate the statistics in Eq. 12 from the training data, in the continuous case (*e.g.* text-conditional) we need to find those statistics also for conditions that were not seen during training. To this end, we first consider a joint space for training samples $\{x_{1_i}, c_i\}_{i=1}^m$, which represents the semantic distances between conditions $c_i$ and samples $x_{1_i}$. In the setting where $c_i$ is text, we choose a pretrained CLIP embedding. $c_i$ is then mapped to this representation space, and then mapped to the data space (could be a latent representation), using a learned mapper $\mathcal{P}_\theta$. Specifically, $\mathcal{P}_\theta$ is trained to minimize the objective:

$$\mathcal{L}_{\text{prior}}(\theta) = \mathbb{E}_{q(x_1, c)}\|\mathcal{P}_\theta(E(c)) - x_1\|_2^2. \tag{14}$$

where $E$ is the pre-trained mapping to the joint condition-sample space (e.g. CLIP). $\mathcal{P}_\theta$ can be seen as approximating $\mathbb{E}[x_1|c]$, which is used as the mean for the condition specific Gaussian. At inference, where new conditions (e.g., texts) may appear, we first encode the condition $c_i$ to the joint representation space (e.g., CLIP) followed by $\mathcal{P}_\theta$. This mapping provides us with the center $\mu_i$ of each Gaussian. We also define $\Sigma_i = \sigma_i^2 I$ where $\sigma_i$ is a hyper-parameter, ablated in Sec. 5.2.1

### 4.3 Training and Inference

Given the prior $p_0$ (either using the data statistics or by training $\mathcal{P}_\theta$), for each condition $c$, we have its associated Gaussian parameters $\mu_c$ and $\Sigma_c$. The map $\psi_t(x|x_1, c)$ must be defined in order to minimize Eq. 10 above. This corresponds to the interpolating maps between this Gaussian at $t = 0$ and a small Gaussian around $x_1$ at $t = 1$, defined by:

$$\psi_t(x|x_1, c) = \sigma_t(x_1, c)x + \mu_t(x_1, c), \tag{15}$$

$$\sigma_t(x_1, c) = t(\sigma_{\min}\mathrm{I}) + (1 - t)\Sigma_c^{1/2}, \quad \text{and} \tag{16}$$

$$\mu_t(x_1, c) = tx_1 + (1 - t)\mu_c. \tag{17}$$

This results in the following target flow vector field

$$u_t(\psi_t(x|x_1, c)) = \frac{d}{dt}\psi_t(x|x_1, c) = \left(\sigma_{\min}\mathrm{I} - \Sigma_c^{1/2}\right)x + x_1 - \mu_c.$$

During inference we are given a condition $c$ and want to sample from $q(x_1|c)$. Similarly to the training, we sample $x_0 \sim p(x_0|c)$ and solve the ODE

$$\frac{d}{dt}\psi_t(x) = v_\theta\left(t, \psi_t(x), c\right), \quad \psi_0(x) = x_0 \tag{18}$$

Training and implementation details are provided in Appendix E.

## 5 Experiments

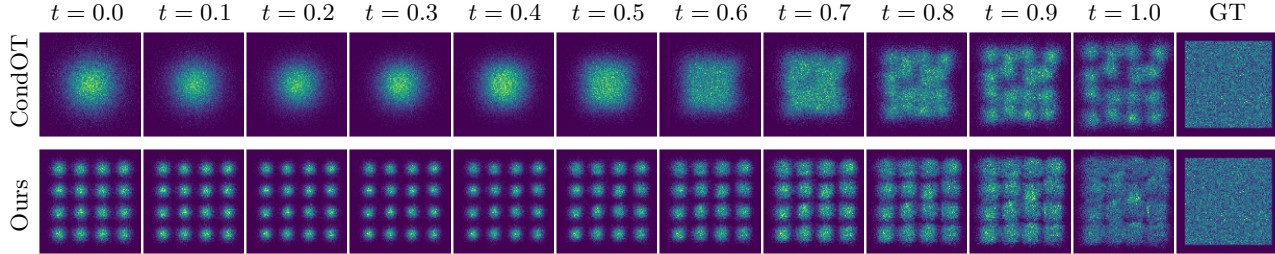

Figure 2: **Trajectory illustration.** A toy example illustrating the trajectory from the source to the target distribution for our method and conditional flow matching using optimal transport (CondOT).

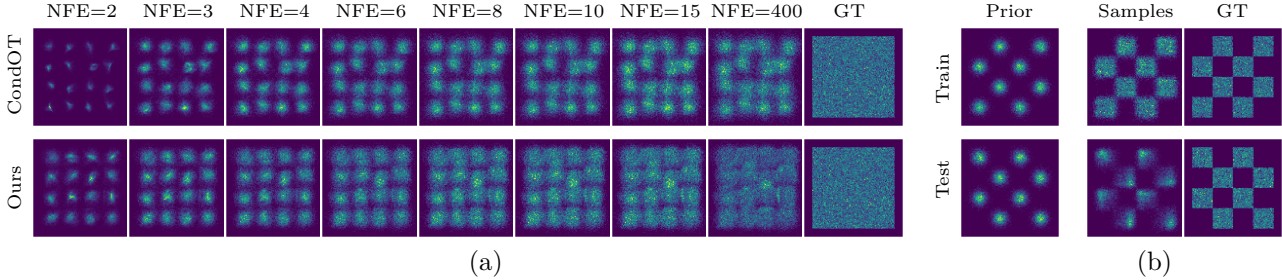

(a)               (b)

Figure 3: (a) **NFE convergence illustration.** A toy example illustrating convergence to the target distribution at different NFEs, for our method, compared to CondOT. (b) **Generalization illustration.** A toy example illustrating the generalization capabilities. LHS: Source prior and target samples for training classes RHS: As for LHS, but for test classes.

**Overview.** This section empirically evaluates our method across both synthetic and real-world settings. Our goal is to demonstrate that initializing the generative process from a structured conditional prior leads to faster convergence, improved numerical efficiency, and higher-quality samples. We begin in Sec. 5.1 with controlled 2D toy experiments that illustrate the intuition behind our approach and its advantages in

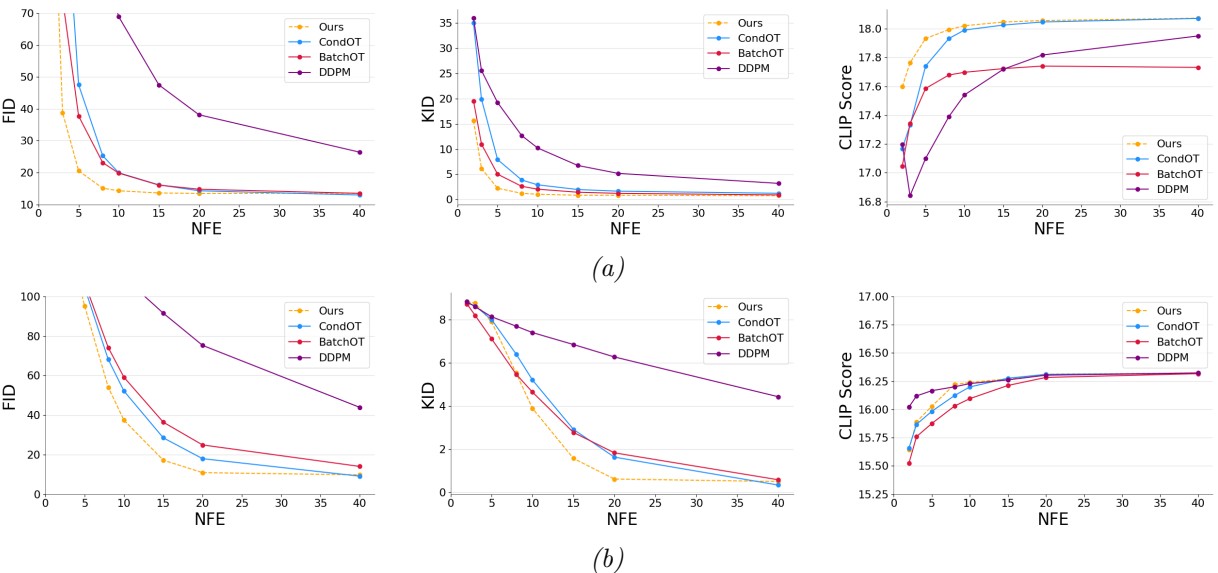

Figure 4: **Numerical evaluation.** *(a)* We compare our method to class conditional flow matching using optimal transport paths (CondOT) (27), BatchOT (33), and DDPM (17), on the ImageNet-64 dataset. We consider the FID score (LHS), KID score (Middle) and CLIP score (RHS). *(b)*. As in *(a)* but for text-to-image generation on the MS-COCO dataset. As can be seen our method exhibit significant improvement per NFE, especially for low NFEs. For example, for 15 NFEs, on ImageNet-64 and MS-COCO we get **FID of 13.62** and **FID of 18.05** respectively, while baselines do not surpass FID of 16.10 and FID of 28.32 respectively for the same NFEs. We consider up to 40 NFE steps and note that DDPM converges to a superior result given more steps.

convergence and generalization. We then turn to real-world image generation tasks in both class-conditioned (ImageNet-64) and text-conditioned (MS-COCO) settings, where we quantitatively compare against existing flow- and diffusion-based baselines (Sec. 5.2.1). Finally, we analyze training efficiency, provide qualitative visualizations, and conduct ablation studies to isolate the contribution of key components of our method.

## 5.1 Toy Examples

We begin by considering the setting in which the prior distribution is a mixture of isotropic Gaussians (GMM), where each Gaussian's mean represents the center of a class (we set the standard deviation to 0.2). The target distribution consists of 2D squares with the same center as the Gaussian's mean in the source distribution and with a width and height of 0.2, representing a large square. We compare our method to class-conditional flow matching (with OT paths), where each conditional sample can be generated from each Gaussian in the prior distribution.

In Fig. 2, we consider the trajectory from the prior to the target distribution. By starting from a more informative conditional prior, our method converges more quickly and results in a better fitting of the target distribution. In Fig. 3(a), we consider the resulting samples for the different NFEs. NFE indicates the number of function evaluation is used using a discrete Euler solver. Our method better aligns with the target distribution with fewer number of steps.

In Fig. 3(b), we consider the model's ability to generalize to new classes not seen during training, akin to text-to-image generation's setting. Training on only a subset of the classes, our model exhibits generalization to new classes at test time, demonstrating that incorporating a conditional prior does not impair the generalization behavior observed in vanilla Flow Matching.

In Fig. 5, we evaluate the method in the case where classes are not uni-modal and there are intersections in the prior distribution, following data from VLines of the Datasaurus Dozen (12). We present generated samples from a model trained using CondOT (*a*) alongside samples from our model (*b, c*).

The maximum mean discrepancy (MMD) computed on this data is 0.084 for CondOT, while we achieve an improved MMD of 0.072. In this setting, the class centers coincide, and the observed improvement arises from differences in the covariance of the condition-specific priors, which provide a better geometric alignment with the data distribution. In addition, we consider the case where the GMM consists of 2 Gaussians for class A and 3 Gaussians for B (as shown in $d$ and $e$

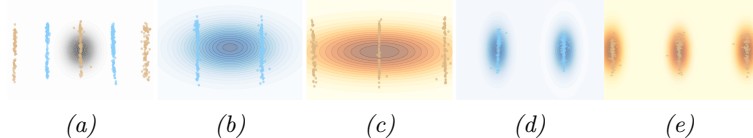

*(a)*       *(b)*       *(c)*       *(d)*       *(e)*

Figure 5: **Multi-modal classes.** 1. A toy example illustrating multi-modal classes with intersections in the prior. Each color represents a class (class A or B), with samples as points and the prior distribution as contour lines. *(a)* shows a standard Gaussian prior (in black), while *(b)* and *(c)* show class-specific priors. While the mean each class falls on samples from the other class, our method results in an improved MMD score. (e) and (d) are as in (b) and (c) respectively, but where a separate GMM is used for each class.

respectively). These Gaussians can be found with a per-class clustering method (applied on training data) such as GMM-based clustering. This results in improved performance compared to using a single Gaussian per class (MMD of 0.067 vs 0.072).

## 5.2 Real World Setting

### Datasets and Latent Representation Space.

For class-conditioned setting, we consider the ImageNet-64 dataset (8) while for text-to-image setting, we consider the 2017 split of the MS-COCO dataset (26), using standard train/validation/test splits. We compute all our metrics on the ImageNet-64 validation set and the MS-COCO validation set. We perform flow matching in the latent representation of a pre-trained variational auto-encoder (42).

### 5.2.1 Quantitative Results

For a fair comparison, we evaluate our method in comparison to baselines using the same architecture, training scheme, and latent representation, as detailed above. We compare our method to standard class-conditioned or text-conditioned flow matching with OT paths (27) which we denote CondOT, where the source distribution is chosen to be a standard Gaussian. We also consider BatchOT (33), which constructed a prior distribution by utilizing the dynamic optimal transport (OT) formulation across mini-batches during training. Lastly, we consider Denoising Diffusion Probabilistic Models (DDPM) (16). We adapted the baselines DDPM, BatchOT/CondOT for conditional generation by incorporating condition $c$ into their shared U-Net architecture via standard mechanisms like cross-attention (detailed in Appendix E). This architectural conditioning is orthogonal to our novel prior contribution. By using identical architectures and conditioning for all methods, our experimental validation fairly shows our prior enhances state-of-the-art conditional generation approaches, improving performance and convergence. To evaluate image quality, we consider the KID (3) and FID (14) scores. We also consider the CLIP score to evaluate the alignment of generated images to the input text or class, using the standard setting, as in (13).

**Overall Performance.** We evaluate the FID, KID and CLIP similarity metrics for various NFE values (as defined above), which is indicative of the sampling speed. In Fig. 4(a) and Fig. 4(b), we perform this evaluation for our method and the baseline methods, for ImageNet-64 (class conditioned generation) and for MS-COCO (text-to-image generation), respectively. As can be seen, our method obtains superior results across all scores for both ImageNet-64 and MS-COCO. For ImageNet-64, already, at 15 NFEs our method achieves almost full convergence, whereas baseline methods achieve such convergence at much higher NFEs. This is especially true for FID, where our method converges at 15 NFEs, and baseline methods only achieve such performance at 30 NFEs. A similar behavior occurs for MS-COCO at 20 NFEs. We note that when considering NFEs for MS-COCO, we consider the pass in the mapper $\mathcal{P}_\theta$ to be marginal due to the small size of the mapper in relation to the velocity $v_\theta$, see Appendix B. In Appendix B, we also compare exclusively the diversity of generated samples in comparison to baselines, demonstrating improved performance.

**Training Convergence Speed.** By starting from our conditional prior distribution, training paths are on average shorter, and so our method should also converge more quickly at training. To evaluate this, in Fig. 6, we consider the FID obtained at each epoch as well as the number of function evaluations (NFE) required for an adaptive solver to reach a pre-defined numerical tolerance, for a model trained on MS-COCO. Specifically, FID is computed using an Euler sampler with a constant number of function evaluations, NFE=20. As for the adaptive sampler, we use the `dopri5` sampler with `atol=rtol=1e-5` from the `torchdiffeq` (5) library. Our method results in lower NFEs and superior FID, for every training epoch.

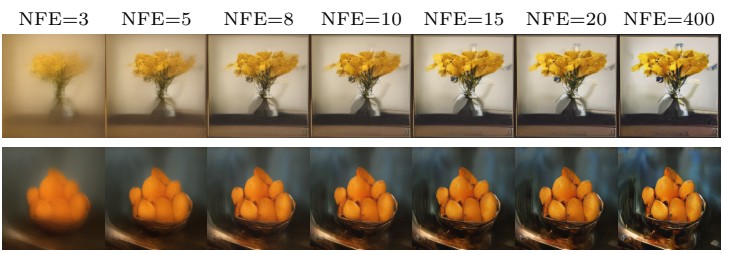

Figure 6: **Training time.** For a text-conditional model trained on MS-COCO, we consider the NFE per training epoch. We compare our method with CondOT, BatchOT and DDPM. Note that DDPM had an FID value above 30 for all epochs so not shown on the RHS.

**Qualitative Results.** In Appendix C, Fig. 9, we provide a visualization of our results for a model trained on MS-COCO, demonstrating both the sample corresponding the the text in the conditional source distribution, which resembles 'an average' image corresponding to the text, as well as samples corresponding to the text. We also provide a diverse set of images generated by our method, in comparison to flow matching. IN Fig. 7, we consider, for a model trained on MS-COCO and a specific prompt, a visualization of our results for different NFEs, illustrating the sample quality for varying numbers of sampling steps. As can be seen, our method already produces highly realistic samples at NFE=15.

Figure 7: **Visualization of results for different NFEs**. We consider a model trained on MS-COCO, and two different validation prompts: Top: "There are yellow flowers inside a vase", Bottom: "A bowl full of oranges".

**Ablation Study.** In the continuous setting, as in MS-COCO, our method requires choosing the hyperparameter $\sigma$, the standard deviation of each Gaussian. In Tab. 2, we report the FID, KID, and CLIP similarity values for different values of $\sigma$. Our method results in best performance when $\sigma = 0.7$, we believe that a relative large $\sigma$ is necessary to allow a richer conditional prior due to the complex nature of the conditional image distribution. We also consider the case where our mapper $\mathcal{P}_\theta$ takes as input a bag-of-words encoding instead of a CLIP encoding showing the importance of an expressive condition representation. As can be seen, performance drops significantly. As an additional ablation, we compare CondOT to our approach when mapping distributions directly to the original data space. When sampling with 15 NFEs, CondOT has an FID score of 27.32 vs our **22.55** on ImageNet-64.

Table 2: **Ablation study.** Model performance for different values of $\sigma$ (the standard deviation) as a hyperparameter for a model trained on MS-COCO. We also consider the case where our mapper $\mathcal{P}_\theta$ takes as input a bag-of-words encoding instead of a CLIP.

|  | FID $\downarrow$ | KID$\downarrow$ | CLIP$\uparrow$ |
|---|---|---|---|
| $\sigma = 0.2$ | 23.55 | 2.88 | **16.12** |
| $\sigma = 0.5$ | 15.47 | 0.93 | 15.75 |
| $\sigma = 0.7$ | **7.55** | **0.61** | 15.85 |
| $\sigma = 1.0$ | 7.87 | 1.66 | 15.81 |
| w/o CLIP | 16.33 | 2.38 | 15.51 |

## 6 Conclusion

In this work, we introduce a novel initialization for flow-based generative models using condition-specific priors, improving both training time and inference efficiency. Our method allows for significantly shorter probability paths, reducing the global truncation error. Our approach achieves improved performance on MS-COCO and ImageNet-64, surpassing baselines in FID, KID, and CLIP scores, particularly at lower NFEs. The flexibility of our method opens avenues for further exploration of other conditional initialization. While this work we assumed a GMM structure of the prior distribution, different structures can be explored.

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

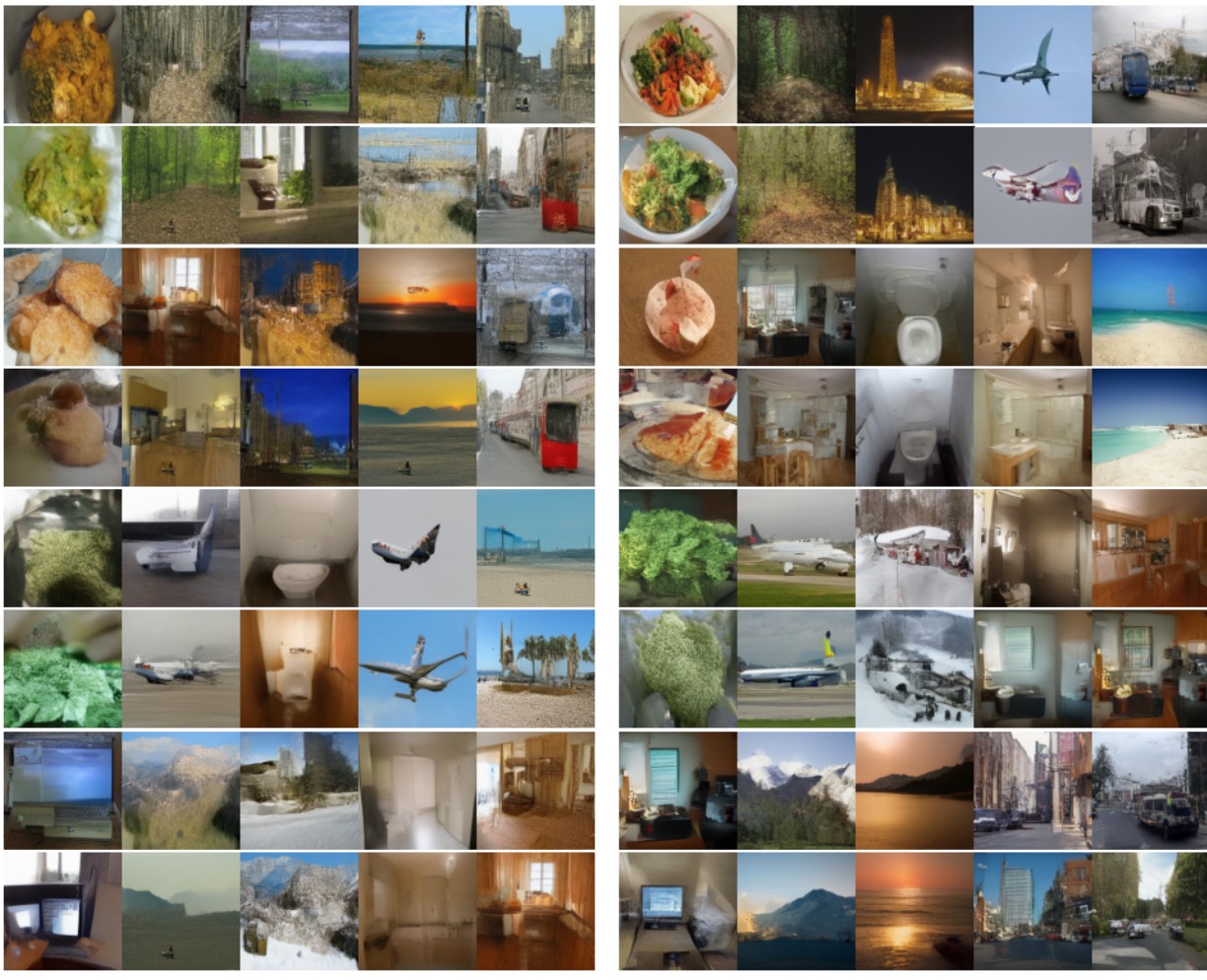

|  |  |
|---|---|
| Flow Matching | Ours |

Figure 8: Visual comparison of randomly generated samples for prompts from the MS-COCO validation set using our method, in comparison to flow matching, for a model trained on MS-COCO.

# A    Theoretical Justification for Lower Truncation Error

We follow our definition of the global truncation error in Eq. 13 and formalize why we expect it to decrease as the prior distribution moves closer to the data. This connection can be formalized under the reasonable assumption that the velocity fields $u_s$ learned via flow matching scale proportionally with the effective distance $D$ between the source and target distributions. Empirically, this is supported by faster convergence with our closer prior (e.g., Fig. 3, Fig. 6). Let $u_s = Du'_s$, where $u'_s$ is a velocity field with magnitude $O(1)$. As argued in Sec. 4.2, the Lipschitz constant $L$ of $u_s$ also scales proportionally: $L = DL'$.

**Proposition 1.** *Assuming flow matching velocity fields $u_s$ have magnitude and Lipschitz constant $L$ scaling proportionally with $D$, then the global truncation error bound decreases as $D$ decreases.*

*Proof.* The error is bounded by $|e_n| <= \frac{max_j \tau_j}{hL} * (e^{L(t_n - t_0)} - 1)$. Assuming $max_j \tau_j$ also scales with $D$, and substituting $L = DL'$, the bound becomes approximately $C(e^{DL'(t_n - t_0)} - 1)$. The exponential dependence on $D$ via $L$ shows that reducing $D$ (closer prior) exponentially reduces the error bound, allowing larger steps $h$ or lower error. $\square$

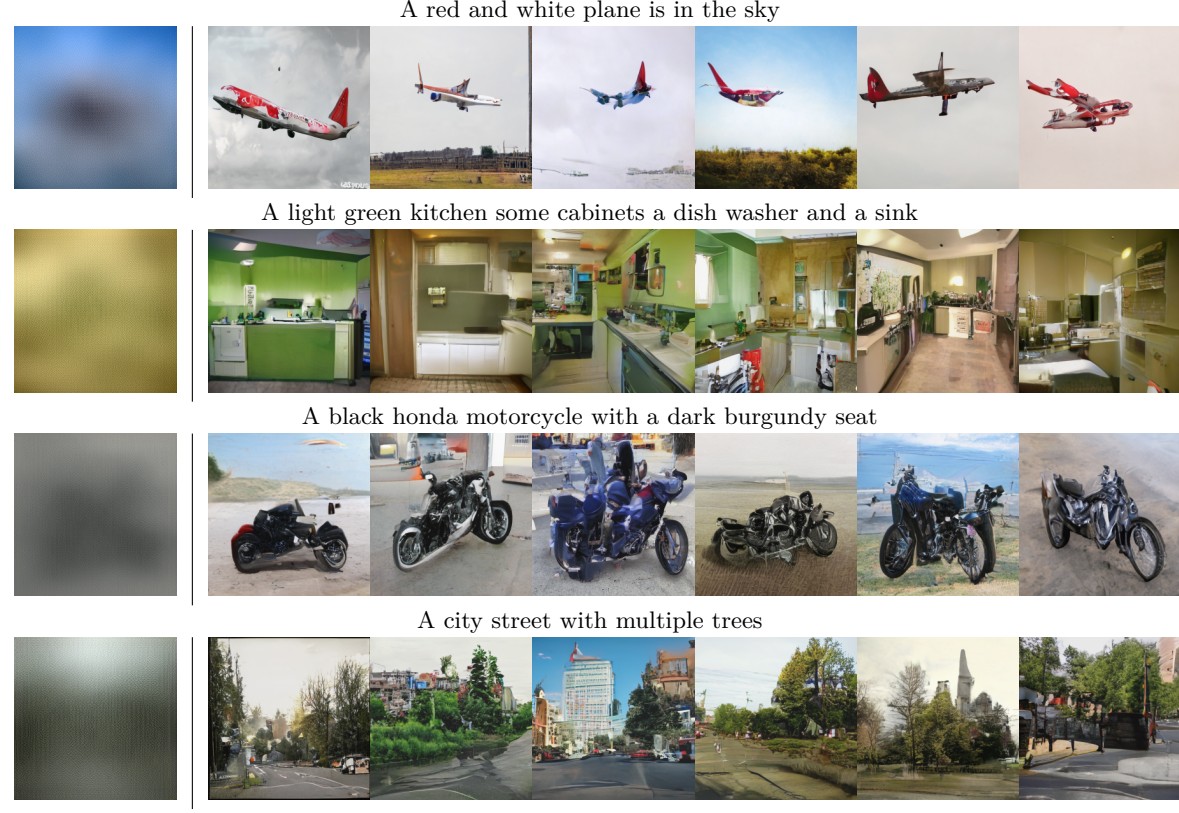

Figure 9: **A visualization of our results on MS-COCO.** We show, for four different text prompts: (a). The sample corresponding to the text in the conditional source distribution, which is used as the center of Gaussian corresponding to the text prompt (LHS) (b). Six randomly generated samples from the learned target distribution conditioned on the text prompt (RHS).

## B  Additional Quantitative Results

In Tab.3, we present additional metrics (FID, KID, and CLIP-Score) for ImageNet-64 with 15 NFEs. We compare the performance of CondOT (27), BatchOT (33) and DDPM (16). As shown, our model delivers significant improvements over the baselines.

As additional comparison, we note that while FID has a component addressing diversity, it measures both fidelity and diversity together. To this end, we considered the Recall and Coverage metrics from (31), which exclusively measure diversity (higher is better for both). On ImageNet-64, compared to CondOT and BatchOT, for 15 NFEs we obtain Recall/Coverage of 0.54/0.78 while CondOT obtains 0.03/0.07 and BatchOT obtains 0.14/0.23. For 40 NFEs, we obtain Recall/Coverage of 0.55/0.79 while CondOT obtains 0.32/0.49 and BatchOT obtains 0.37/0.60. This demonstrates that our method improves the generated distribution's diversity, in both low and high NFEs.

Table 3: **Numerical evaluation.** Quality of generated samples (FID, KID), and conditional fidelity (CLIP-Score) for our method in comparison to baselines, for the ImageNet-64 dataset for 15 NFEs. We consider (CondOT) (27), BatchOT (33) and DDPM (16).

|         | FID ↓     | KID ↓    | CLIP ↑    |
|---------|-----------|----------|-----------|
| DDPM    | 47.51     | 6.74     | 17.71     |
| CondOT  | 16.16     | 1.96     | 18.02     |
| BatchOT | 16.10     | 1.43     | 17.72     |
| Ours    | **13.62** | **0.83** | **18.05** |

## C  Visual Results

In Fig. 9, we provide a visualization of our results for a model trained on MS-COCO. We show, for four different text prompts: (a). The sample corresponding to the text in the conditional source distribution,

Table 4: Hyper-parameters used for training each model

|  | ImageNet-64 | MS-COCO |
| --- | --- | --- |
| Dropout | 0.0 | 0.0 |
| Effective Batch size | 2048 | 128 |
| GPUs | 4 | 4 |
| Epochs | 100 | 50 |
| Learning Rate | 1e-4 | 1e-4 |
| Learning Rate Scheduler | Constant | Constant |

which is used as the center of Gaussian corresponding to the text prompt. (b). Six randomly generated samples from the learned target distribution conditioned on the text prompt. As can be seen, the conditional source distribution samples resemble 'an average' image corresponding to the text, while generated samples display diversity and realism.

In Fig. 7, we consider, for a model trained on MS-COCO and a specific prompt, a visualization of our results for different NFEs, illustrating the sample quality for varying numbers of sampling steps. As can be seen, our method already produces highly realistic samples at NFE=15.

In Fig. 8, we provide additional visual results for our method in comparison to standard flow matching for a model trained on MS-COCO.

## D  Additional Toy Examples

We construct two synthetic target distributions that share the *same* conditional prior to illustrate the behavior of our method when the prior statistics are identical. The first is a "+" shape composed of intersecting horizontal and vertical line segments, while the second is obtained by rotating this distribution by $45°$ (Fig. 10). Both distributions share an isotropic Gaussian conditional prior $p(x_0|c) = \mathcal{N}(0, I)$, providing a controlled setting for analyzing equivalence and divergence regimes of our approach.

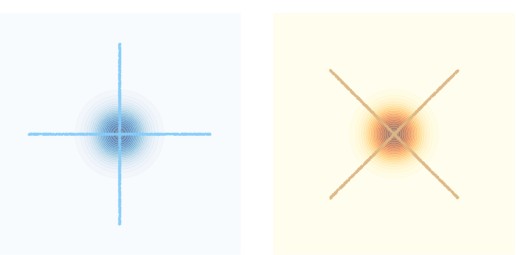

Figure 10: **Toy distributions with identical conditional priors.** (*Left*) A "+" shape formed by intersecting horizontal and vertical line segments. (*Right*) The same distribution rotated by $45°$. Both share an isotropic Gaussian conditional prior centered at the origin.

When both the prior and data distributions are isotropic, our method effectively reduces to standard Flow Matching (CondOT), producing comparable flow trajectories and sample quality. However, when the data variance deviates from one, the structured prior yields measurable advantages. In this case, we observe improved alignment between the prior and data distributions, with an MMD of 0.077 compared to 0.082 for the vanilla baseline (CondOT).

## E  Implementation Details

We report the hyper-parameters used in Table 4. All models were trained using the Adam optimizer (23) with the following parameters: $\beta_1 = 0.9$, $\beta_2 = 0.999$, weight decay $= 0.0$, and $\epsilon = 1e−8$. All methods we trained (*i.e.* Ours, CondOT, BatchOT, DDPM) using identical architectures, specifically, the standard Unet (36) architecture from the `diffusers` (44) library with the same number of parameters (872M) for the the same number of Epochs (see Table 4 for details). For all methods and datasets, we utilize a pre-trained Auto-Encoder (42) and perform the flow/diffusion in its latent space.

In the case of text-to-image generation, we encode the text prompt using a pre-trained CLIP network and pass to the velocity $v_\theta$ using the standard UNet condition mechanism. In the class-conditional setting, we

create the prompt 'an image of a $\langle class \rangle$' and use it for the same conditioning scheme as in text conditional generation.

For the mapper $\mathcal{P}_\theta$ (Sec. 4.2), we use a network composed of a linear layer followed by two ResNet blocks, totaling 11M parameters. The training time required for $\mathcal{P}_\theta$ is negligible compared to the flow network, accounting for only $\sim 1\%$ of the total training time.

When using an adaptive step size sampler, we use `dopri5` with `atol=rtol=1e-5` from the `torchdiffeq` (5) library.

Regarding the toy example Sec. 5.1, we use a 4 layer MLP with ReLU activation as the velocity $v_\theta$. In this setup, we incorporate the condition by using positional embedding (43) on the mean of each conditional mode and pass it to the velocity $v_\theta$ by concatenating it to its input.

## F  Limitations and Future Work

Our proposed approach, utilizing a Gaussian Mixture Model (GMM) prior centered on mapped conditional means within a VAE latent space, demonstrates significant improvements for class-conditional and text-to-image generation tasks, proving effective in these complex, real-world settings.

**Choice of Gaussian prior.**   In this work, we intentionally model each conditional prior as a Gaussian distribution, which fully captures the first two moments (mean and covariance) of the conditional data. This design ensures analytical simplicity, stable training, and computational efficiency, while already yielding substantial improvements in transport efficiency and generative quality. Nevertheless, richer or learnable priors could offer additional flexibility. For example, mixture-based or neural priors that capture higher-order moments may further reduce the transport distance or better adapt to multi-modal condition distributions. Exploring such priors—either learned jointly with the flow model or parameterized via amortized inference—remains an important direction for future work.

**Beyond image domains.**   While our approach proves effective for image generation, other conditional tasks involve fundamentally different output structures for which the current GMM-based prior may be suboptimal. For example, in conditional 3D human pose generation, data lie on a complex kinematic manifold, and an "average pose" used to center a Gaussian component might be physically invalid. More generally, structured modalities such as human motion, molecular graphs, or music exhibit strong geometric, relational, or temporal dependencies that are not well captured by Gaussian components. Adapting our framework to these settings may require priors that are manifold-aware or that explicitly model structural constraints—such as distributions defined over joint angles, graph topologies, or temporal sequences. Developing such task-specific conditional priors constitutes a promising direction for future work. Finally, investigating performance directly in the data space, without relying on a pretrained VAE encoder, also represents an interesting path for improving fidelity and interpretability.

# G Training and Inference Algorithms

In this section, we provide concise pseudocode for the main components of our method: ($i$) training the flow model from a conditional prior (CGJFM), ($ii$) inference (sampling) from the learned model, and ($iii$) training the prior mapper $\mathcal{P}_\theta$ for continuous conditioning (e.g., text).

**Notation.** $\psi_t(x_0|x_1, c)$ and $\dot{\psi}_t(x_0|x_1, c)$ denote the interpolation and its time derivative, respectively, as defined in Sec. 4. $\mathcal{N}(\mu, \Sigma)$ denotes a Gaussian distribution with mean $\mu$ and covariance $\Sigma$. The loss $\mathcal{L}_{\text{cgjfm}}$ is given in Eq. 10.

---

**Algorithm 1 Training Flow Matching from a Conditional Prior (CGJFM)**

---

**Require:** Data $\{x_1, c\}$, conditional prior $\{\mu_c, \Sigma_c\}$, flow network $v_\theta$
1: **while** not converged **do**
2:   Sample $(x_1, c) \sim q(x_1, c)$, $t \sim \mathcal{U}(0, 1)$, $x_0 \sim \mathcal{N}(\mu_c, \Sigma_c)$
3:   $x_t \leftarrow \psi_t(x_0|x_1, c)$, $u_t \leftarrow \dot{\psi}_t(x_0|x_1, c)$
4:   $\mathcal{L}_{\text{cgjfm}}(\theta) = \|v_\theta(t, x_t, c) - u_t\|^2$
5:   $\theta \leftarrow \theta - \eta \nabla_\theta \mathcal{L}_{\text{cgjfm}}$
6: **end while**

---

**Algorithm 2 Sampling from the Conditional Prior**

---

**Require:** Condition $c$, trained flow $v_\theta$, prior $(\mu_c, \Sigma_c)$
1: Sample $x_0 \sim \mathcal{N}(\mu_c, \Sigma_c)$
2: **for** $t \in [0, 1]$ **do**                                                    ▷ ODE integration
3:   $\dot{x}_t = v_\theta(t, x_t, c)$
4: **end for**
5: **return** $x_1$

---

**Algorithm 3 Training $\mathcal{P}_\theta$ for Continuous Conditions**

---

**Require:** Data $\{x_1, c\}$, encoder $E$, network $\mathcal{P}_\theta$
1: **while** not converged **do**
2:   Sample $(x_1, c) \sim q(x_1, c)$
3:   $\hat{\mu}_c \leftarrow \mathcal{P}_\theta(E(c))$
4:   $\mathcal{L}_{\text{prior}}(\theta) = \|\hat{\mu}_c - x_1\|^2$
5:   $\theta \leftarrow \theta - \eta \nabla_\theta \mathcal{L}_{\text{prior}}$
6: **end while**

---

These pseudocode listings correspond directly to the procedures described in Sec. 4. Algorithm 1 trains the conditional flow network $v_\theta$ using our conditional prior distribution, Algorithm 2 performs sampling by integrating the learned ODE, and Algorithm 3 trains $\mathcal{P}_\theta$ to map conditions (e.g., CLIP embeddings) to the mean of the condition-specific Gaussian prior.

