# OpenReview forum: "Designing a Conditional Prior Distribution for Flow-Based Generative Models"
_TMLR — Accepted by TMLR_

### Review · Reviewer_cD6g · 2025-09-12

**Summary Of Contributions:**

### Summary
This paper proposes a novel improvement of flow-based generative models to utilize condition-specific prior distributions. To be specific, the paper highlights that existing flow-based models employ a unimodal distribution to generate the target distribution, which typically involves a large flow distance from the source noise to the target point, resulting in more sampling steps and longer training times. To overcome this limitation, the authors focus on utilizing a conditional prior distribution in a conditional generative modeling method, specifically flow matching. Specifically, the authors describe how this concept can be implemented with a particular family of conditional prior distributions, namely the Gaussian Mixture Model. Using experiments on toy datasets and two real-world datasets, the authors demonstrate how the proposed method enhances overall training performance.
### Strengths
1. The concept of conditional prior distribution is well-motivated.
2. Generally, the paper writing is clear and easy to understand.
3. The experimental results support the improvement well.
### Weaknesses
1. I don’t find any major flaws or weaknesses in this paper. There are some questions or minor changes listed in the **Requested Changes** section.

**Additional Comments:**

This paper is well-written in general. The use of conditional prior distributions is a novel contribution to flow-based generative models. The overall writing is clear and easy to understand. Also, the experimental results support the improvement well.

**Audience:**

Yes

**Audience Explanation:**

To the best of my knowledge, the flow-based generative model is also one of the mainstream generative models in machine learning research. The underlying geometric nature of the normalizing flow maps each point in the prior distribution to some other point in the posterior distribution. Considering the flow of each point during sample generation, some readers may think that the topological/geometric mismatches between the prior and posterior distributions are cumbersome, and they would be interested in reducing these geometric gaps.

**Broader Impact Concerns:**

I don’t see a particular broader impact concern regarding this paper.

**Claims And Evidence:**

Yes

**Claims Explanation:**

The main claim of the paper is that the use of conditional prior distributions can reduce both the number of samplings required during test time and overall training time, resulting in high-quality samples with more efficient training. The authors quantified the quality of the generated samples using three different metrics: FID, KID, and CLIP. They successfully demonstrated that the proposed method can converge faster than other existing methods without conditional prior distributions.

**Requested Changes:**

1. At the beginning of Section 5, please describe the main structure of the experimental section, with a summary of the goal of the experiments.
2. I’m not quite convinced by the result in Figure 3(b). Do the authors have evidence (or experimental results) that the same result cannot be achieved unless conditional prior distributions are utilized?
3. Figure 5 presents the case where the centers are the same, showing improvements in the maximum mean discrepancy. Do the authors claim that the improvement still comes from selecting class-specific priors with different covariances?
4. I’m curious about the case when there are two target distributions whose conditional prior distributions are the same. For example, a uniform distribution over the union of horizontal and vertical line segments (+ shape) would have an isotropic Gaussian distribution as a conditional prior. Rotating this distribution by 45 degrees would result in the same conditional prior. Then, we can construct two target distributions with identical conditional prior distributions. Will the proposed method not have any improvement, because it will be the same as the existing methods? If there are still some improvements from the proposed method, why?

---

> ### Author Response · Authors · 2025-10-08
>
> We are very pleased that the reviewer found our work novel, clearly written, and well supported by experimental evidence. We greatly appreciate the positive evaluation highlighting the strength of our motivation, the clarity of exposition, and the demonstrated improvements in efficiency and sample quality. We also thank the reviewer for the thoughtful and constructive suggestions, which we have addressed comprehensively in the revised version as detailed below.
>
> $\textbf{Addition of experiment overview (Requested Change 1):}$
> As suggested, we have added a paragraph at the beginning of the experimental section summarizing its structure and goals. The new text reads:
>
> Overview.
> This section empirically evaluates our method across both synthetic and real-world settings. Our goal is to demonstrate that initializing the generative process from a structured conditional prior leads to faster convergence, improved numerical efficiency, and higher-quality samples. We begin in Sec. 5.1 with controlled 2D toy experiments that illustrate the intuition behind our approach and its advantages in convergence and generalization. We then turn to real-world image generation tasks in both class-conditioned (ImageNet-64) and text-conditioned (MS-COCO) settings, where we quantitatively compare against existing flow- and diffusion-based baselines (Sec. 5.2). Finally, we analyze training efficiency, provide qualitative visualizations, and conduct ablation studies to isolate the contribution of key components of our method.
>
> This addition clarifies the logical flow of the experiments and the motivation behind each stage.
>
>
> $\textbf{Clarification of Figure 3(b) (Requested Change 2):}$
> The purpose of this experiment is to demonstrate that incorporating a conditional prior does not degrade the model’s generalization capabilities compared to vanilla Flow Matching (FM). As shown in Sec.~5.1, our method maintains comparable performance on unseen classes, confirming that the structured initialization preserves the desirable generalization behavior of standard FM while providing efficiency gains in convergence and sampling.
> This point is further clarified and discussed in the revised manuscript (see Sec 5.1).
>
>
> $\textbf{Clarification of Figure 5 (Requested Change 3):}$
> Yes, the improvement persists even when the class centers coincide. In this case, the conditional priors differ primarily in their covariances, which still provide a better geometric match to the data distributions. As a result, the transport distance is reduced, leading to measurable gains in convergence and sample quality.
> This clarification has been explicitly added to the revised manuscript for clarity.
>
>
> $\textbf{On identical conditional priors (Requested Change 4):}$
> When two target distributions share identical conditional priors and have unit variance, our method effectively reduces to standard flow matching (CondOT), as expected.
> However, when the data variance deviates from one, the structured prior still offers benefits. In our experiments, we observed improved alignment (e.g., MMD = 0.077 vs 0.082 for the vanilla baseline), confirming that our approach remains advantageous in this case.
> We further discuss this scenario and provide corresponding visualizations in the revised manuscript (see Appendix D).

---

### Review · Reviewer_2Pv1 · 2025-09-12

**Summary Of Contributions:**

This paper proposes a novel method for improving the efficiency and performance of conditional flow-based generative models. The core idea is to replace the standard, generic, unimodal prior distribution (e.g., a standard Gaussian) with an informative, condition-specific prior.  The authors validate this approach on toy examples, ImageNet-64, and MS-COCO, demonstrating promixing improvements in FID, KID, and CLIP alignment scores compared to baselines, particularly at low sampling steps.

**Audience:**

Yes

**Audience Explanation:**

The central contribution—designing a conditional prior to shorten the transport path—is well-motivated, and intuitive. It addresses a clear limitation in existing conditional flow and diffusion models.

**Claims And Evidence:**

Yes

**Claims Explanation:**

- The experimental results are convincing. The method shows promising quantitative improvements over baselines (CondOT, BatchOT, DDPM) across multiple metrics (FID, KID, CLIP) and datasets (ImageNet-64, MS-COCO). Specifically, the model performs at a low number of function evaluations (NFE) for same quality, as shown in Figure 4.

- The paper is well written, and the experiments are done thoroughly to support the claim. The authors also include ablations studies for in-depth analysis of the proposed approach.

**Requested Changes:**

- As stated in Sec 5.1, the prior is modeled as an isotropic Gaussian. This feels like a significant simplification. The performance might be further improved by learning a more complex prior. The paper would be strengthened by a discussion of this design choice and its potential limitations.

- The proposed method is only tested on ImageNet-64 dataset and MS-COCO dataset. It's better to evaluate on more complex real-world tasks such as video generation, 3D generation to validate the generalization ability.

---

> ### Author Response · Authors · 2025-10-08
>
> We are delighted that the reviewer found our work well-motivated, clearly written, and experimentally convincing. We greatly appreciate the positive evaluation emphasizing the strength of our experimental validation, the clarity of presentation, and the potential impact of introducing condition-specific priors for efficient flow-based generation. We also thank the reviewer for the thoughtful suggestions, which we have addressed in the revised manuscript as detailed below.
>
> $\textbf{Choice of Gaussian prior:}$
> We intentionally model the conditional prior as an isotropic Gaussian to maintain analytical simplicity and computational efficiency, while fully capturing the first two moments (mean and covariance) of the conditional data.
> This formulation ensures stable optimization and provides a clean analytical form that integrates naturally with the flow-matching objective. However, we acknowledge that this assumption also introduces limitations: a Gaussian prior cannot capture higher-order correlations or multimodality that may exist within complex conditional distributions, potentially limiting expressivity in highly structured data domains. Richer priors-such as mixture-based or neural priors that model higher-order moments-could better represent these distributions and may further improve performance at the cost of additional computational complexity.
> We have added a detailed discussion of this design choice, its advantages, and its limitations in the revised manuscript (see Appendix F).
>
>
> $\textbf{Scope of experiments:}$
> Following standard practice in the literature, we evaluate our method on large-scale image datasets (ImageNet-64 and MS-COCO), which serve as canonical continuous-signal benchmarks for conditional generative modeling. This choice allows for direct comparison with prior flow-based and diffusion methods. While extending our approach to more complex modalities, such as video or 3D generation, is indeed compelling, such experiments require substantially greater computational resources. We therefore leave this exploration to future work.

---

### Review · Reviewer_iUYo · 2025-09-26

**Summary Of Contributions:**

This paper proposes to use a prior distribution for flow-based generative models, which enables the reduction of computational costs. The authors introduce two methods for designing the prior distribution using Gaussian mixture models, one for a class-labeled dataset and the other for text prompts. The experimental results demonstrate that the proposed method can improve training times and generation efficiency compared to baseline methods.

[Strengths]
- Introducing the prior distribution for flow-based generative models is well-motivated and interesting.
- The proposed method experimentally works well to reduce the training and generation times and improve the generation quality.
- The proposed method can be combined with various types of flow-based generative models and other techniques.

[Weaknesses]
- The proposed method requires additional computational cost for constructing the prior distribution, although it is not significant compared to the flow-network training cost as described in the Appendix. However, to introduce the prior distribution, the full training of the flow network is required.

**Audience:**

Yes

**Audience Explanation:**

The generation cost of flow-based generative models is a central issue in the field. As the proposed method directly contributes to reducing this cost, it will likely interest researchers and developers working on generative models and their applications.

**Broader Impact Concerns:**

I do not have concerns about the ethical implications of this paper.

**Claims And Evidence:**

Yes

**Claims Explanation:**

The motivation of the proposed method is clearly described. The use of the prior distribution for reducing the sampling steps of generation is convincing. The effectiveness of the proposed method is demonstrated through experiments on the toy dataset and the ImageNet and MS-COCO datasets.

**Requested Changes:**

I think this paper is well-written. I do not have any critical issues or major concerns.
The following are some (minor) suggestions.

- It might be better to provide the training and inference procedures of the proposed method as pseudocode, which would help readers understand the method in further detail.
- It would be nice to release the code for the proposed method.
- I cannot identify the exact architectures of $P_\theta (E(c))$. Could you please explain the detailed implementation? Especially, in my understanding, $\mathcal{P}_\theta$ maps the encoded representation (continuous vector space) to the image space.

---

> ### Author Response · Authors · 2025-10-08
>
> We are very encouraged that the reviewer found our work well-motivated, clearly presented, and experimentally convincing. We appreciate the positive assessment highlighting both the novelty of introducing a conditional prior for flow-based generative models and its demonstrated efficiency gains. We also thank the reviewer for the thoughtful minor suggestions, which we have addressed in the revised version as detailed below:
>
> $\textbf{Training and inference procedures:}$
> To improve clarity, we have added pseudocode outlining both the training and inference stages of our method in Appendix G. This addition provides a clearer overview of the proposed pipeline.
>
> $\textbf{Code release:}$
> Upon acceptance, we will release the full implementation upon acceptance to ensure reproducibility and facilitate further research.
>
> $\textbf{Clarification on $P_\theta$​:}$
> The network $P_\theta$​ maps the encoded representation (e.g., CLIP embedding) to the image domain, which may correspond to either pixel space or a VAE latent space, depending on the specific experimental setup. Detailed pseudocode is provided in Appendix G for reference.

---

> > ### Comment · Reviewer_iUYo · 2025-10-09
> >
> > Thank you for the response.
> > I have no more questions or concerns.
> >
> > Best,

---

### Decision · Action_Editor_hY6q · 2025-11-06

**Recommendation:** Accept as is

**Audience:**

Yes

**Audience Explanation:**

The paper will be of interest to researchers working on generative models.

**Claims And Evidence:**

Yes

**Claims Explanation:**

The paper considers the prior in flow-based generative models. The idea is to replace a unimodal Gaussian distribution with a Gaussian Mixture based on the conditioning random variable. For discrete conditioning, the Gaussians are extracted from the conditional data distributions. The authors also extend this idea to text conditioning. The method shows promising results: faster training, fewer sampling steps requited, and higher quality generations. The method is validated empirically on toy and real-world tasks.

The claims of the paper are well-supported by the evidence. The experiments demonstrate an improvement from the proposed method convincingly.

The reviewers unanimously support the paper.